# Assessing the aesthetic attractivity of European butterflies: A web-based survey protocol

Elia van Tongeren[1,2], Ginevra Sistri[2], Vincenzo Zingaro[3], Alessandro Cini[4], Leonardo Dapporto[1,2], Mariagrazia Portera[3] *

1 NBFC, National Biodiversity Future Center, Palermo, Italy, 2 Department of Biology, University of Florence, Florence, Italy, 3 Department of Humanities and Philosophy, University of Florence, Florence, Italy, 4 Department of Biology, University of Pisa, Pisa, Italy

☯ These authors contributed equally to this work.
* mariagrazia.portera@unifi.it

**Editor:** Łukasz Kajtoch, Institute of Systematics and Evolution of Animals Polish Academy of Sciences, POLAND

**Data Availability Statement:** The first publication in a scientific journal resulting from the data of this experiment will be led by the coordinators of this

## Abstract

Aesthetic attractivity stands as an underestimated yet fundamental feature of species in conservation biology, significantly driving disproportionate protection efforts towards charismatic species. Despite the evidence, few attempts sought to precisely quantify the impact of aesthetic attractivity in defining priority of species for conservation actions (e.g. inclusion in International Union for Conservation of Nature red lists and protection lists). This study protocol describes the setting of an online test (available from April 2022 to April 2023 at www.unveiling.eu) designed to i) quantify the aesthetic attractivity to humans of the 496 European butterfly species and ii) identify which features (both in the perceived animal and in the perceiver) influence the aesthetic attractivity of a given butterfly species. The test is divided in 5 sections (personal data, ranking, single morphological features, emotional engagement, dispositional variables) aimed at profiling the relation each participant has with the species examined. In the long-term, evaluating butterflies' aesthetic attractivity could facilitate the critical assessment of current conservation strategies, such as the process of selection of flag and umbrella species by research institutions, environmental associations and Non Governative Organizations. This is expected to provide the much-needed evidence to set up unbiased biodiversity conservation strategies and counteract the selective anthropogenic pressure which favours the extinction of unattractive species, being no or less protected compared to charismatic species.

## Introduction

Aesthetic values play a substantial role in almost every aspect of human everyday experience [1,2]. This is the case even with domains, apparently more objective and less obviously influenced by human aesthetic choices, such as scientific disciplines (mathematics, physics, biology etc.; see [3–5]). With specific reference to conservation biology, it has been argued that processes such as the choice of the subjects of interest by conservationists, the choice of

initiative. The data will be made publicly available in an open access repository along with the first publication, so anyone will be able to access the dataset and make additional use of it. The data set containing the test results, completed by the participants up to the time of writing, used for the preliminary calculations is available at https://github.com/leondap/files/blob/main/unveiling.zip.

**Funding:** Our work has been supported mainly by five sources of funding: 1. MP and LD have been awarded a University (national) grant for peer-reviewed excellent projects, with selection procedure by the Academic Senate of the University of Florence (URL: https://www.letterefilosofia.unifi.it/index.html?newlang=eng) held in November 2020. The review process was carried out by a commission of 10 anonymous external reviewers selected by the MIUR (Ministry of Education, University and Research) and by CINECA (Inter-university Consortium of the North-East for Automatic Calculation). These reviewers were selected from the REPRISE (Digital Register of Scientific Experts for the Scientific Evaluation of Italian Research) database. The project was evaluated according to the following criteria: Excellence of the research project (based on the coherence with "Horizon 2020" themes, clarity and relevance of the objectives, soundness of the idea, progress beyond the state of the art, potential for innovation and ambition, credibility of the proposed approach), impact of the research project, quality and efficiency of the implementation of the research project. The funding was granted to MP and LD for the project "Unveiling" relating respectively to the Department of Letters and Philosophy and the Department of Biology. The total amount of funding granted is € 42,205.00. 2. On September 30 2021, the Academic Senate of the University of Florence (URL: https://www.letterefilosofia.unifi.it/index.html?newlang=eng) approved the research project "Smart Beauty. Theory and practice of the role of the aesthetic dimension in the strategies of conservation of endangered species", P.I. MP, co-P.I. LD. The project has been financed with a total amount of 150,000 euros gross coming from PNRR (Piano Nazionale di Ripresa e Resilienza) Ministerial funds (URL: https://italiadomani.gov.it/en/home.html). VZ has been appointed through a selection procedure held in Fall 2021 to the position of fixed-term researcher covered by the funds assigned to the project. 3. EvT and GS have been awarded research grants for the conservation and monitoring of pollinators funded by national park bodies following the Habitats Directive of the Ministry of Ecological Transaction (URL: https://www.mite.gov.it/). The funds have been provided by the

representative animal species for raising-awareness projects by NGOs, and even the allocation of public funds for research initiatives are influenced by aesthetic values [6–15]. As a result, in the last few years the study of aesthetics has started to emerge as a key topic in conservation, as witnessed by a growing amount of research over very wide branches of the tree of life [16–25].

It is indeed well known that some groups like birds [16,19,20], coral reef fishes [17] and big mammals [26] are generally recognized by humans as natural beauties. For this reason, they are considered ambassadors of biodiversity (flag species [27,28]) and harnessed by environmental organisations like WWF to gain public support for their campaigns and to motivate people to invest resources in conservation. The same scenario applies to butterflies which constitute a marked exception within insects. Indeed, while insects are generally not considered as popular and charismatic animals [29,30], an extraordinary aesthetic merit is attributed to butterflies (intended as superfamily Papilionoidea) [14,31,32]. This is likely due to their striking and charming colours and forms, their diurnal activity, making us more accustomed to encounters with them, to their "friendly" appearance (butterflies are generally perceived as harmless) and to their increasingly recognized role as pollinators.

Europe hosts 496 butterflies species, whose huge morphological variability in wing patterns, shapes and colours, due to mimetic, thermal and sexual strategies, provides a rich substrate for aesthetic attractivity. Indeed, in the history of Western aesthetics, features such as variation, novelty, extravagance have been traditionally understood as highly aesthetic [33–35], as recognized by Charles Darwin himself in his attempt at making sense of the human and non-human aesthetic dimension within the evolutionary framework [36–38].

In conservation biology, functional features such as those related to morphology (e.g. body size), feeding (e.g. ingestion rate), life history (e.g. reproduction mode), physiology (e.g. temperature tolerance), behaviour (e.g. dispersal mode) [39] represent the key-features determining a species' fitness and survival in a given environment. Arguably, each species' aesthetic attractivity to humans constitute a still underappreciated yet fundamental feature driving disproportionate conservation efforts towards charismatic species [14] and facilitating their persistence in highly human-impacted landscapes ("anthropogenic selection", [40]). Butterflies seem to suit this framing [14].

Yet, only few studies [41–43] have attempted to design protocols to rigorously quantify the aesthetic attractivity (less precisely, "beauty") of butterflies and its potential impact on conservation policies. For this reason, the "Unveiling" research project, led by the University of Florence (Italy), aims to test the hypothesis that aesthetic attractivity to humans increases the chances of survival of endangered butterfly species. In this paper, we describe the protocol of an online test (available at www.unveiling.eu) designed to i) quantify the aesthetic attractivity of European butterflies to humans; ii) identify which features (both in the perceived animal and in the perceiver) influence the aesthetic attractivity of a given butterfly species.

In the long-term, evaluating butterflies' aesthetic attractivity will provide the much-needed evidence to set up unbiased biodiversity conservation strategies and counteract, therefore, the selective anthropogenic extinction of unattractive species [40,44].

## Materials and methods

The first aim of our study is to assess aesthetic attractivity of 496 European butterfly species. Previous studies have mostly addressed the topic of aesthetic attractivity "per se" [17–20], i.e. tracing it back to properties in the animal (as "causes" of the perceived attractivity). In our perspective, the aesthetic attractivity of a species should be considered as a relational functional feature, resulting from the interaction between the item/object and the human perceiver. What humans aesthetically like is indeed deeply influenced by what they feel, know, are

project "Ricerca e conservazione sui lepidotteri diurni di sei Parchi Nazionali dell'Appennino centro-settentrionale" and by the project of the Parco Nazionale dell'Arcipelago Toscano named "Ricerca e conservazione sugli Impollinatori dell'Arcipelago Toscano e divulgazione sui Lepidotteri del parco" within the Direttiva Biodiversità 2019-2020 of the Italian Ministry of Ecological Transaction. 4. Part of the dissemination activity has been funded by the project awarded to LD "Ricerca, divulgazione e protezione dei Lepidotteri nella riserva mondiale della Biosfera UNESCO del Monte Peglia, ai fini di favorire meccanismi di resilienza climatica dei principali ecosistemi della riserva" PSR Umbria 2014-2020 - D.D. n. 6572/2019 (URL: https://www.montepegliaperunesco.it/). 5. EvT and LD acknowledge the support of NBFC to University of Florence, Department of Biology, funded by the Italian Ministry of University and Research, PNRR, Missione 4 Componente 2, "Dalla ricerca all'impresa", Investimento 1.4, Project CN00000033. The funders had no role in study design, data collection and analysis, decision to publish, or preparation of the manuscript.

**Competing interests:** The authors have declared that no competing interests exist.

interested in [45]. For this reason, we first evaluated i) the importance of some butterfly species' morphological features in influencing the perceived attractivity, and then considered ii) the type and extent of emotional engagement elicited by butterflies' images in the survey participants and iii) the participants' dispositions and interests towards natural sciences and aesthetics and the arts.

The test, available for mobile and desktop devices at the link www.unveiling.eu, consists of 6 different parts which are thoroughly described below (to see the full list of questions see S1 Appendix) and summarised in Fig 1.

Before starting the test, the participant is presented with an introductory page which provides some preliminary information such as: test duration (about 10 minutes); presence of a timer indicating that a limited amount of time is available for some answers; invitation to spontaneously answer the questions without thinking too much over them; declaration that no personal data are collected (in accordance to the European Regulation 2018/1725); the reward provided to the participant at the end of the test (butterfly pdf guides, reading suggestions); butterfly photo credits (available for download) based on iNaturalist dataset. After these introductory remarks, the participant starts the test, which is divided into five different sections.

The Ethics Committee for Research of the University of Florence with the opinion n. 168 of July 9, 2021 approved the general structure of the project "Identification, evaluation and application potential of the aesthetic dimension in conservation practices: butterflies as case study, between new theoretical ideas and practical applications" and the methods of execution of the same as illustrated by the scientific responsible Mariagrazia Portera and therefore expressed positive opinion on the project.

Furthermore, considering the extent of the project and the fact that no sensitive personal data are collected, the Ethics Committee deemed it appropriate not to require informed consent from participants, be they also under 18 years of age.

## Section n. 1: Personal data

It has been shown that factors such as gender, level of education, cultural background, age etc. can substantially influence our aesthetic preferences, appreciation, liking and disliking [46–49]. The first section comprises questions aimed at collecting general participant data anonymously: gender (male, female, non-binary), age, nationality, level of studies, employment sector, colour blindness. This section, together with the sections about interests and dispositions (see below), is expected to contribute to unveil the role of personal features in the aesthetic appreciation of endangered animals.

## Section n. 2: Ranking

In this section participants are presented with a panel of 9 pictures of different butterfly species, when the user clicks on each picture to rank it, they see that picture enlarged (covering the other previews) with a clickable 0–10 meter to express their choice and they are asked to attribute a score to each of them in response to the question "How beautiful do they look to you?". For each species we have collected up to 8 images of living individuals in their natural environment, representing all the combinations of: i) sex (male or female), ii) dorsal or ventral view, iii) flower or neutral backgrounds. For monomorphic species, whose sexual morphological differences are indistinguishable in pictures, only four pictures were collected. In some cases it has not been possible to obtain 8 pictures per species. For example, some species rarely visit flowers (e.g. *Charaxes jasius*) making it impossible to take pictures of them on flowers. Many lateral basking butterflies do not open their wings while resting, so photos of the species' dorsal view were unavailable (e.g. *Gonepteryx* spp.). The poor contactability of a few rare

## PERSONAL DATA

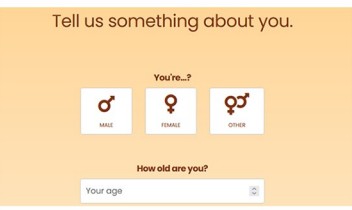

- ☞ Anonymous collection of personal data such as gender, level of education, cultural background, age.

- ☞ This section contribute to unveil the role of personal features in the aesthetic appreciation of different animal species.

## RANKING

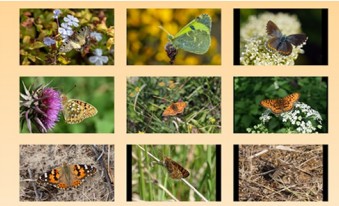

- ☞ Participants are asked to attribute a score (from 1 to 10) to 9 pictures of butterflies according to their aesthetic preferences.

- ☞ Collection of aesthetic appreciation data computes three different indices for each butterfly species.

## MORPHOLOGICAL FEATURES

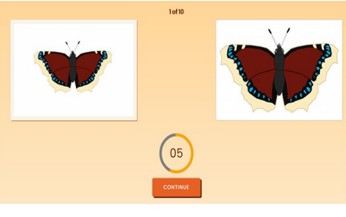

- ☞ Participants are presented with pairs of butterfly drawings with natural aspect and morphological alterations.

- ☞ Importance of specific morphological features for participant aesthetic preferences.

## EMOTIONAL ENGAGEMENT

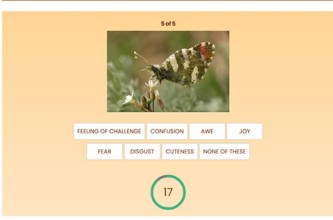

- ☞ Participants are asked to identify the emotion aroused by each picture and its intensity.

- ☞ Evaluation of the link between emotions and aesthetic experience.

## DISPOSITIONAL VARIABLES

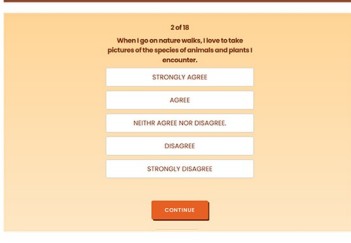

- ☞ Collection of data about the participants interests and inclinations towards natural science and the arts.

- ☞ This section retrieves data about the role of personal dispositions in the aesthetic appreciation of different animal species.

**Fig 1. "Unveiling" test.** In this figure are presented the 5 sections of the online "Unveiling" test. For each section a screenshot, a brief description and the summary of objectives are provided.

species made it impossible to gather all 8 images. The list of butterfly species along with the collected pictures is provided in S1 Table.

To ensure a similar number of tests per species, the probability of a species to be selected is inversely proportional to the times it has been chosen in previous tests. This is done by

attributing a random number between 0 and 1 to each species which was summed to the standardised number of times each species was chosen (times chosen divided by maximum number of times chosen in the dataset). For each test, each species potentially ranged from a minimum of 0 to a maximum of 2, and we selected the 9 species showing the lowest values. Once the species are selected, the choice between the pictures referring to that species is random.

No time limit applies to this section of the test; the participant can linger on each image as long as they want, also zooming on each photo to appreciate every detail before providing their score.

The pictures have been obtained by iNaturalist citizen science platform, selecting only the images with a Creative Commons (CC) licence, and the author photo credits list is available for download in the first introductory section of the test. Furthermore, for a little set of species pictures, not available as CC in iNaturalist platform, we directly asked the authors permission. The following websites also provided significant iconographic material: www.farfalleitalia.it; www.leps.it; www.pyrgus.de; www.europebutterflies.com; www.flickr.com; www.lepidoptera.eu; www.lepiforum.org; http://kajsnatur.dk.

## Section n. 3: Single morphological features

In this section, 10 pairs of butterfly-drawings created *ad hoc* are shown to each participant in random positions (left-right). A 10 seconds' countdown timer is displayed in order to invite the participant to express their preference for one of the two images quickly and instinctively. In each pair, one drawing faithfully represents the morphological-perceptual key-features of a particular species of butterfly, while the other represents the same key-features but altered. As for the features, on the basis of a substantial body of research both in empirical aesthetics and in conservation biology [45,50–54], we selected: i) butterfly dimension, ii) colours of the wings contrast intensity, iii) grouping and order of the design patterns of the wings, iv) forewing/hindwing proportion, v) presence or absence of wing eyespots, vi) wing eyespots dimensions, vii) presence or absence of wing tails, viii) wing tail length, ix) smooth or jagged wing edges. In each pair of drawings, every alteration of the aforementioned traits comes from artificial interventions (through MS PowerPoint, Windows vers. 17) meant to highlight the aesthetically relevant features and to comparatively identify their influence on the votes given by the test-takers. The altered butterfly-drawings have no reference whatsoever to real phenotypes.

The species reproduced in the drawings (see Table 1) have been chosen because of the high perceptual perspicuity of the key-features. An example is shown in Fig 2, where two *Aglais io* drawings are compared, one of them presenting eyespots in the wings (natural aspect, Fig 2A), the other one lacking them (modified butterfly aspect, Fig 2B). All the couples of drawings with respective features analysis are available in S2 Appendix.

In this table are the single morphological features analysed in the third section of the "Unveiling" test, butterfly species associated with them and number of cases for each feature according to the classification shown in S2 Appendix.

## Section n. 4: Emotional engagement

In this section the participant's emotional engagement is investigated. Emotions significantly affect our aesthetic experiences [55–57]. In order to assess the role of emotions in butterfly aesthetic attractivity, we present the participant with a selection of 5 images of butterflies and we ask them to identify the emotion that comes closest to what they feel, also quantifying the emotional intensity in a scale range from 1 to 10. The species selected for this section of the test are representative both in terms of morphological characteristics and phylogenetic diversity, in

**Table 1. Single morphological features analysed in the third section of the "Unveiling" test.**

| Morphological features | Butterfly species | Number of cases |
|---|---|---|
| Butterfly dimension | *Nymphalis antiopa* | 2 (natural, 36% smaller) |
| Colours of the wings contrast intensity | *Charaxes jasius* | 3 (natural, 40% brighter and 0% contrast, 40% less bright and 40% more contrast) |
| Grouping and order of the design patterns of the wings | *Charaxes jasius; Erebia medusa* | 2 (natural, modified with an unordered and random arrangement of wing elements) |
| Forewing / hindwing proportion | *Kirinia roxelana* | 2 (natural, modified with altered fore wing / hind wing proportion) |
| Presence or absence of wings eyespots | *Aglais io; Erebia medusa* | 2 (natural, modified aspect without wing eyespots) |
| Wings eyespots dimensions | *Aglais io; Erebia medusa* | 3; 2 (natural, modified aspect with 100% smaller (only for *A. io*) eyespots, modified aspect with 100% bigger eyespots) |
| Presence or absence of wings tails | *Iphiclides podalirius; Charaxes jasius* | 2 (natural, modified aspect without wing tails) |
| Wings tails length | *Iphiclides podalirius; Charaxes jasius* | 2 (natural, modified with 20% longer wing tails for *I. podalirius* and 15% longer wing tails for *C. jasius*) |
| Smooth or jagged wings edges | *Polygonia c-album* | 3 (natural, modified with jagged edges, modified with smooth edges) |

order to ensure a heterogeneous selection of butterflies that can elicit a wide range of emotions in the participants, based on both peculiar traits and outward appearance of species belonging to different phyla. Relying on previous studies in the field of empirical aesthetics [45,58–60], we selected the following emotions: awe, confusion, joy, disgust, fear, cuteness, feeling of challenge or none of these [61–63]. This section is time-constrained: the participant is invited to make their choices in 20 seconds, 10 to express their emotion and 10 to quantify it. As for the species represented in each selection of five pictures, we have selected 26 species representative of 5 families of European butterflies: 5 Papilionidae (*Iphiclides podalirius*, *Papilio machaon*, *Parnassius apollo*, *Archon apollinus*, *Zerynthia cassandra*), 5 Pieridae (*Gonepteryx cleopatra*,

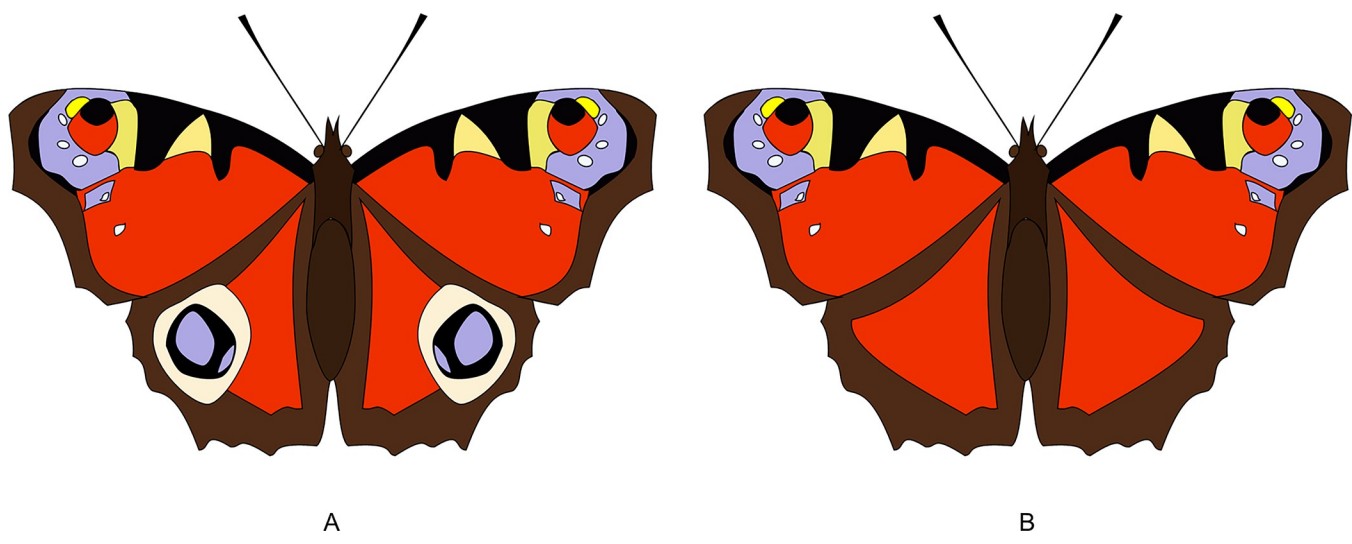

A                                                                                       B

**Fig 2. Two *Aglais io* drawings.** (A) One butterfly drawing is with eyespots (natural aspect) (B) and the other is without eyespots (modified aspect) on posterior wings.

*Colias hyale, Aporia crataegi, Anthocharis cardamines, Leptidea sinapis*), 5 Lycaenidae (*Lycaena dispar, Callophrys rubi, Satyrium w-album, Agriades orbitulus, Polyommatus icarus*), 6 Nymphalidae (*Issoria lathonia, Argynnis paphia, Aglais io, Charaxes jasius, Polygonia c-album, Coenonympha oedippus*), 5 Hesperiidae *(Heteropterus morpheus, Carterocephalus silvicola, Ochlodes sylvanus, Spialia therapne, Pyrgus sidae*).

The species listed above cover only the first 2,500 tests that will be carried out. After the first 2,500 answers, the 26 pictures will be substituted by another cluster of 25 pictures portraying the following species (representative of all the 6 families of European butterflies):

2 Papilionidae (*Parnassius mnemosyne, Iphiclides feisthamelii*), 4 Pieridae (*Pieris rapae, Colias alfacariensis, Zegris pyrothoe, Leptidea reali*), 4 Lycaenidae (*Phengaris arion, Cupido minimus, Lysandra coridon, Polyommatus thersites*), 1 Riodinidae (*Hamearis lucina*), 11 Nymphalidae (*Libythea celtis, Limenitis reducta, Boloria euphrosyne, Euphydryas cynthia, Apatura ilia, Danaus chrysippus, Melanargia galathea, Hipparchia semele, Erebia cassioides, Maniola jurtina, Erebia tyndarus*), 3 Hesperiidae (*Gegenes pumilo, Gegenes nostrodamus, Carcharodus alceae*). This change is meant to grant a higher degree of representativity of the butterfly species diversity (as to subfamilies and tribes).

## Section n. 5: Dispositional variables. Interests and inclinations

Being interested into and favourably predisposed towards the natural sciences, on the one hand, and the arts and aesthetic experiences on the other hand are among the most crucial individual differences which influence people's response to nature and the environment, and to beauty and the aesthetic [19,20,64–69]. In this section, we collect data about the participants' dispositions towards natural science and the arts in general. We present the participant with a questionnaire which is a modified version of a model (about subjective dispositions towards the arts) administered and validated by Chamorro-Premuzic & Furnham [70]. To reduce the fatigue effect [71,72], we shortened the original questionnaire from 36 to 18 questions.

The participant is offered a 5-point Likert scale (strongly agree; agree; neither agree nor disagree; disagree; strongly disagree) from which to select a response. In compliance with the Likert scale standard protocol, the 18 questions are asked half in a negative form, half in a positive one; the order of positive and negative questions randomly occurs, so as to avoid possible acquiescence effects [73,74].

Our test is freely available in both Italian and English language on our dedicated website www.unveiling.eu. The test has been online since April 2022 and can be accessed without limitations of age, nationality, education and socio-cultural background. So far, the "Unveiling" test has been advertised by means of the "snowballing" technique [75], i.e. through national magazines in Italy, direct mailing to researchers' contacts as well as social media, asking the participants to forward the test to their own families and contacts. Moreover, it has been promoted through a wide array of public engagement events in Italy (museums, libraries, university events, edutainment and citizen science events, during "bioblitz" in Italian National Parks etc.). The main target of the test are European citizens, in line with the biodiversity conservation strategies our project is focused on. To obtain a sample size able to support reliable analyses, we plan to involve at least 5,000 participants, corresponding approximately to 100 scores for each species in the ranking section. The test will presumably be online until April 2023.

## What outcomes will be measured, when and how

**Section n. 2: Ranking.** Single species mean scores obtained in the "ranking" section of the test will provide a first index of the aesthetic attractivity. Such an index is expected to be strongly variable among pictures and less accurate for those species with high sexual

dimorphism and strong differences between dorsal/ventral sides. For these reasons we will compute a second index, only considering the mean scores of the sex/side of each species reaching the highest scores in the test. A source of possible uncertainty in the results is the disposition of participants to provide higher or lower scores. For this reason we will calculate a third index as the mean scores scaled and centred among those of the same participant. These three different indexes (species mean scores, mean scores of the best aspect [sex and dorsal/ventral wing view], mean scaled scores) will be submitted to a PCA (Principal Component Analysis) in order to extract an expected single factor, accounting for the combination of these three indexes. This factor can be considered as the aesthetic attractivity value, which will be tested for accuracy by using two intrinsic features of butterflies:

1. Since highly phylogenetically related butterfly species are more similar to each other, we expect to find a strongly significant phylogenetic signal in the aesthetic attractivity value. This will be tested using the recently published time-calibrated phylogenetic tree of European butterflies [76] and a typical test for phylogenetic signal, the phylosig function of the phytools R package [77].

2. European butterfly cryptic taxa, objectively identified by Voda et al. [78], should obtain more similar scores to each other than random pairs.

It is reasonable to suppose that picture quality, in particular different backgrounds, may affect the score assigned by people to butterflies. To evaluate the magnitude of this effect we calculated average scores for each species for pictures with and without flower on the background separately. Then we applied a mixed generalised linear model (using glmmTMB R package) comparing the average scores for pictures with and without a flower on the background using species as a random factor. The effect of the background has been evaluated in two ways: 1) by obtaining estimated marginal means for the background factor by using the emmeans R package 2) by obtaining using the r.squaredGLMM function of the 'MuMIn' package the proportion of variance explained by different backgrounds (marginal $R^2$) and that explained by the full model including the species random factor (conditional $R^2$).

**Section n. 3: Single morphological features.**   The preference for either the natural or the altered version of different butterfly features will be assessed by using a Chi-squared test. We will also evaluate the possible interaction between the participant's preference for a specific version of a given feature and their dispositional variables. For example, participants self-reporting higher-than-average knowledge in butterflies and strong interest in natural sciences might prefer the drawing portraying the non-altered version of the species.

Then, it will be possible to score the presence and/or the magnitude of these features in the European butterflies (e.g. with presence-absence variables or with more complex morphological analyses) and to verify which of them most explain the aesthetic attractivity value (section "ranking") by a phylogenetic regression [79] using aesthetic attractivity value as a response variable, the scores for morphological features of each species as predictors while correcting for phylogenetic autocorrelation using the phylogenetic tree of European butterflies [76].

**Section n. 4: Emotional engagement.**   In this section we will provide the analysis of the data concerning the emotional engagement of the participants. Since representatives for all butterfly subfamilies will be included in this section, we will perform a chi-squared test to assess whether images of phylogenetically distant species of butterflies with different morphological features are associated with different elicited emotions, while phylogenetically close species should arouse similar emotional responses. For example, we would expect paler coloured and smaller butterflies, such as the Lycaenidae family, to be associated with cuteness and lower emotional intensity, whereas brighter coloured and bigger ones (e.g. Papilionidae) should

arouse awe [80–82] and in general a higher intensity. Moreover, we could expect to find an interaction between the self-reported level of knowledge and interest and the kind of emotions elicited: notably, experts should experience fear and/or disgust less frequently [45,83–85] while reporting more frequently emotions with positive valence towards rare or endangered species.

**Section n. 1 and section n. 5: Personal data and dispositional variables.** We will use a PCA (Principal Component Analysis) algorithm (like dudi.mix of the ade4 R package) on personal data and dispositional variables datasets in order to extract the main factors (PCs) associated with specific characteristics of the user. Such principal components will then be used to detect the possible influence of experiences and dispositions of each participant in the answers given to the questions concerning the first three sections of the test (ranking, emotional engagement and single morphological features). As a general rule we expect that dispositional factors show significant interactions in the relationships between species attributes and participant responses in all the sections of the test.

## Preliminary assessment of test appreciation

The first period of beta testing (April 1 2022 to May 22 2022) provided the results for the first 500 participants. Among them, 453 entered the "ranking" section of the test and 94.5% of them ranked all the 9 proposed species; 428 participants entered the "single morphological features" section and 418 of them expressed their preferences to all the 10 pairs of butterfly drawings; 412 participants entered he "emotional engagement" section and 402 of them evaluated all the five proposed species; 376 participants entered the "dispositional variables" section; 361 out of 376 answered all the questions. Overall, about three quarters of the participants went through all the main sections of the test and more than 90% answered all the questions in each section.

The evaluation for the effect of different backgrounds on votes revealed a significant effect with pictures representing a flower in the background receiving higher votes (estimate, 0.165; standard error, 0.0417; z-value, 3.950; p-value <0.001). However, the difference in estimated marginal means in quite low (mean vote without flower 6.42 +- 0.0441 standard error, with flower 6.59 +- 0.0453) and the variance explained by the full model (species+background, conditional $R^2$ = 0.585) is much higher than the variance explained by background alone (residual $R^2$ = 0.007). Nevertheless, due to the significant effect for the presence of a flower in the background we decided to replace all the images at the midpoint of the assessment (more or less 2500 test) so that the effect(s) of any uncontrolled feature(s) of individual pictures is halved. This is one of the means by which we expect to evaluate, in the analyses, the effect of confounding factors (i.e. geographical distribution and the interaction with the living environment of the test-respondents).

Another small yet not negligible impact registered throughout the beta-testing period's results, concerns reported experiences of discomforts caused by the very view of butterfly pictures. Out of 500 finished preliminary tests, in fact, we collected 1823 feedbacks about the emotions experienced by the participants with 160 claims of negative ones, 45 of which reporting "disgust", 37 reporting "fear" and 78 experiencing "confusion" for a total of 8,77% of the whole.

## The status and timeline of the study

We started collecting data on May 1st, 2022. The test will be online until April 2023. April 2021/May 2022

Development and beta testing.

May 2022

Launch of the website.

<u>May 2022/April 2023</u>

Advertisement of the test and dissemination activities.

<u>April 2023</u>

End of data collection and statistical processing of collected data.

<u>September 2023</u>

Publication of results.

## Discussion

In our view, aesthetic attractiveness of different butterfly species to humans represents a feature influencing their possibility to survive in the Anthropocene. A choice of which species to protect influenced by their aesthetic value may not be an optimal strategy to maintain ecosystem functionality. Indeed, charismatic species selected on aesthetic grounds belong to a few phylogenetic clades, thus encompassing a disproportionately low fraction of evolutionary and functional diversity [14,17]. Being such an aesthetic attractivity driven protection undesirable, we need to determine and quantify the aesthetic attractivity of target species in order to evaluate the occurrence of such bias in conservation policies and activities (red lists, funds for establishment of conservation actions).

This is not to overlook that, in other cases, aesthetic attractivity can indeed work as a flywheel and a booster, rather than a brake, for biodiversity protection. Flag and umbrella species are often chosen among the most attractive taxa and their importance in advertising diversity loss is widely known [86].

The protocol described here is designed to determine and quantify the aesthetic index and to analyse its interconnection with different kinds of emotions and with a set of dispositional variables in the human perceiver. The long-term goal of our project is to set up targeted educational strategies to re-modulate people's current (non-functional) aesthetic experience regarding the different species of butterflies, so as to make it more effective for conservation purposes. Our aesthetic experience of the same objects and phenomena can indeed change over time, and aesthetic standards and principles are not set in stone. For instance, we tend to be more attracted by the things we know more, i.e. to get attached to the things we are most frequently exposed to [87–89], although boredom is notoriously a limiting condition of the exposure effect [90]. On the other hand, it is not always true that the better informed our aesthetic judgements are, the stronger or more pleasurable is the aesthetic experience we get [91], since a pivotal role in aesthetic experience is also played by personal engagement, openness and other dispositional variables [92]. In this sense, by providing people with more information about butterflies, by offering them more valuable occasions to interact with the natural world and with insects in particular, and by fostering their interests and openness to nature by means of targeted educational activities [93], it could be possible to re-tune the public aesthetic experience of butterflies towards a more functional set of target species and to take advantage more effectively of the fascination that butterflies exert. Instead of being attracted exclusively by beautiful butterflies, for instance, people could start to care also for unusual, original, surprising, "diverse" forms and colours [94].

### Fitness and expected impact

In conservation biology, many traits are available for European butterflies and they typically describe intrinsic functional traits (such as size, host plants, phenology and behaviour) and

variables referring to habitat preferences (temperature, precipitation, altitude and vegetational units) [39]. These traits have been largely used to predict the decline of species under current environmental changes [95–98].

Adding a new relational aesthetic dimension to biological conservation can facilitate the critical assessment of current conservation strategies, such as the process of selection of certain species (over others) as flag species and umbrella species by research institutions, environmental associations and NGOs and the selection of the species of butterflies to be included into national, international and European conservation actions (e.g. CEE habitat directive, LIFE projects, IUCN red lists, local red list, National and regional protection lists).

## Limitations and improvements of the study design

Our test has so far been disseminated mainly through the snowballing technique, initially including our research group closest contacts (friends, relatives and colleagues) and subsequently expanding the diffusion range to "contacts of contacts". This allows an effective spreading of the test, but at the same time it compromises the representativeness of the sample. Indeed, people involved will be in the majority direct or indirect contact of the researchers and/or more involved in the field of conservation biology and of natural sciences than the general public. Moreover, our study sample includes a large number of participants (at least 5000 planned), but it is still a limited sample in relation to the European population whose representativeness is sought. In fact, it is very difficult to obtain an effective representativeness of all the various segments of the European population (age, geography, sociocultural aspects etc.). That said, however, it should be pointed out that in the case of this study the test will not investigate in detail the population's sociocultural aspects and how these affect the aesthetic appreciation, although the same test could be reproposed and distributed in a more oriented way to study the various sociocultural segments in future research. Another possible limitation of our study is the technological mediation, that is the device through which the participants carry out their aesthetic experience to the vision of the butterflies' images. In fact, the butterfly images presented on the PC, tablet or mobile phone may not accurately reflect the real appreciation of butterflies in nature. Static images (digital reproductions) of dynamic living beings are evaluated [99], while the appreciation in nature would include aspects related to the dynamism of the organisms (e.g. butterfly flight diversity in the four dimensions of space) that are impossible to include in the test. Surely, these are interesting aspects to be considered in future studies, in which we could test the aesthetic appreciation of people towards certain animal species directly in nature or more easily using new technologies such as VR (virtual reality), in order to ensure dynamism and immersivity during the experience [100,101].

## Measures to maximise impact: Dissemination, exploitation and communication of results

At the academic level, the release of scientific papers in high-ranking journals is expected; in line with the Open Access policy, every research product will be freely available online. As the project targets specific grounds for action, public and private institutions and organisations managing protected areas are relevant addressees of the study. Stakeholders as the six national parks of the Central-Northern Italian Apennines and institutions such as the Italian Ministry of Ecological Transition, Butterfly Conservation Europe, W.W.F., Legambiente, ALI (Associazione Lepidotterologica Italiana) will be involved in the design of specifically thought-of events at both specialist and non-specialist level (i.e. bioblitz, guided nature walks, field trips, talks and workshops). These events are expected to take place starting from September 2022 with the discussion of the preliminary results and, from September 2023, with the disclosure of the

final results of the study. Schools and students represent a further target of the dissemination activities; we have planned a series of seminars aimed at engaging students from secondary schools to universities, mainly in the city of Florence (IT) and neighbouring areas. The coverage of the project results will also involve public media such as newspapers, television and social media; regular updates on the main social networks (i.e. Twitter, Facebook and Instagram), along with online videos and podcasts, are also planned.

## Supporting information

**S1 Table. The list of butterfly species pictures.** The list of butterfly species along with the collected pictures used in section n.1 of "Unveiling" test.
(DOCX)

**S1 Appendix. Test question list.** All the questions proposed to the participants during the test are collected together with their answer options in this file.
(DOCX)

**S2 Appendix. Single morphological features drawings.** All the couples of drawings with respective features analysis are available in this file.
(DOCX)

## Acknowledgments

The authors would like to thank Roger Vila, CSIC Research Scientist at the Pompeu Fabra University, Institute of Evolutionary Biology, Barcelona, Spain; Alice Chirico, Research Fellow at the Catholic University of the Sacred Heart, Department of Psychology, Milan, Italy; the colleagues of the University of Florence, Department of Literature and Philosophy, Florence, Italy.

## Author Contributions

**Conceptualization:** Elia van Tongeren, Ginevra Sistri, Alessandro Cini, Leonardo Dapporto, Mariagrazia Portera.

**Data curation:** Elia van Tongeren, Ginevra Sistri, Leonardo Dapporto.

**Formal analysis:** Leonardo Dapporto.

**Funding acquisition:** Leonardo Dapporto, Mariagrazia Portera.

**Investigation:** Elia van Tongeren, Ginevra Sistri, Alessandro Cini, Leonardo Dapporto, Mariagrazia Portera.

**Methodology:** Elia van Tongeren, Ginevra Sistri, Alessandro Cini, Leonardo Dapporto, Mariagrazia Portera.

**Resources:** Elia van Tongeren, Ginevra Sistri.

**Supervision:** Alessandro Cini, Leonardo Dapporto, Mariagrazia Portera.

**Validation:** Alessandro Cini, Leonardo Dapporto, Mariagrazia Portera.

**Visualization:** Elia van Tongeren, Ginevra Sistri.

**Writing – original draft:** Elia van Tongeren, Ginevra Sistri, Alessandro Cini, Leonardo Dapporto, Mariagrazia Portera.

**Writing – review & editing:** Elia van Tongeren, Ginevra Sistri, Vincenzo Zingaro, Alessandro Cini, Leonardo Dapporto, Mariagrazia Portera.

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
