## [Decision Letter · Decision Letter 0]

22 Nov 2022

PONE-D-22-26118Assessing the aesthetic attractivity of European butterflies: a web-based survey protocolPLOS ONE

Dear Dr. Portera,

Thank you for submitting your manuscript to PLOS ONE. After careful consideration, we feel that it has merit but does not fully meet PLOS ONE’s publication criteria as it currently stands. Therefore, we invite you to submit a revised version of the manuscript that addresses the points raised during the review process.

Three Reviewers responded and presented their opinions on this study protocol. These opinions are highly different, with two suggesting only minor revisions and one rejection. However, even these positive reviews include some important issues. Considering the most critical revision I decided to give you the opportunity to respond to the critical comments from all three opinions. I share the opinion that in this protocol is too much subjectivity and that some issues are speculative. I understand that the idea of this protocol had to include some level of subjectivity as it is based on the test, which is filled by people, however, I am also convinced by some of the comments of the Reviewers (particularly this most critic). As this protocol could be a valuable tool for the conservation of butterflies, its publication had to be preceded by careful evaluation of its assumptions, feasibility and potential impact. If you are able to respond to these critical comments and convince Reviewers that your idea for this protocol is right,  I am willing to consider the publication of this protocol.

We look forward to receiving your revised manuscript.

Kind regards,

Łukasz Kajtoch, Ph.D.

Academic Editor

PLOS ONE

Journal Requirements:

“Our work has been supported mainly by four sources of funding:

    1.     MP and LD have been awarded a University (national) grant for peer-reviewed excellent projects, with selection procedure by the Academic Senate of the University of Florence (URL: https://www.letterefilosofia.unifi.it/index.html?newlang=eng) held in November 2020. The review process was carried out by a commission of 10 anonymous external reviewers selected by the MIUR (Ministry of Education, University and Research) and by CINECA (Inter-university Consortium of the North-East for Automatic Calculation). These reviewers were selected from the REPRISE (Digital Register of Scientific Experts for the Scientific Evaluation of Italian Research) database. The project was evaluated according to the following criteria: Excellence of the research project (based on the coherence with “Horizon 2020” themes, clarity and relevance of the objectives, soundness of the idea, progress beyond the state of the art, potential for innovation and ambition, credibility of the proposed approach), impact of the research project, quality and efficiency of the implementation of the research project. The funding was granted to MP and LD for the project "Unveiling" relating respectively to the Department of Letters and Philosophy and the Department of Biology. The total amount of funding granted is € 42,205.00.

    2.     On September 30 2021, the Academic Senate of the University of Florence (URL: https://www.letterefilosofia.unifi.it/index.html?newlang=eng) approved the research project "Smart Beauty. Theory and practice of the role of the aesthetic dimension in the strategies of conservation of endangered species'', P.I. MP, co-P.I. LD. The project has been financed with a total amount of 150,000 euros gross coming from PNRR (Piano Nazionale di Ripresa e Resilienza) Ministerial funds (URL: https://italiadomani.gov.it/en/home.html). VZ has been appointed through a selection procedure held in Fall 2021 to the position of fixed-term researcher covered by the funds assigned to the project.

    3.     EvT and GS have been awarded research grants for the conservation and monitoring of pollinators funded by national park bodies following the Habitats Directive of the Ministry of Ecological Transaction (URL: https://www.mite.gov.it/). The funds have been provided by the project “Ricerca e conservazione sui lepidotteri diurni di sei Parchi Nazionali dell’Appennino centro-settentrionale” and by the project of the Parco Nazionale dell’Arcipelago Toscano named “Ricerca e conservazione sugli Impollinatori dell’Arcipelago Toscano e divulgazione sui Lepidotteri del parco” within the Direttiva Biodiversità 2019-2020 of the Italian Ministry of Ecological Transaction.

      4.    Part of the dissemination activity has been funded by the project awarded to LD “Ricerca, divulgazione e protezione dei Lepidotteri nella riserva mondiale della Biosfera UNESCO del Monte Peglia, ai fini di favorire meccanismi di resilienza climatica dei principali ecosistemi della riserva” PSR Umbria 2014-2020 - D.D. n. 6572/2019 (URL: https://www.montepegliaperunesco.it/).”

7. We note that Figure 1 in your submission contain copyrighted images. All PLOS content is published under the Creative Commons Attribution License (CC BY 4.0), which means that the manuscript, images, and Supporting Information files will be freely available online, and any third party is permitted to access, download, copy, distribute, and use these materials in any way, even commercially, with proper attribution. For more information, see our copyright guidelines: http://journals.plos.org/plosone/s/licenses-and-copyright.

   1. You may seek permission from the original copyright holder of Figure(s) [#] to publish the content specifically under the CC BY 4.0 license.

Reviewers' comments:

Reviewer's Responses to Questions

**Comments to the Author**

1. Does the manuscript provide a valid rationale for the proposed study, with clearly identified and justified research questions?

Reviewer #1: Yes

Reviewer #2: Yes

Reviewer #3: Partly

2. Is the protocol technically sound and planned in a manner that will lead to a meaningful outcome and allow testing the stated hypotheses?

Reviewer #1: Yes

Reviewer #2: Partly

Reviewer #3: Partly

3. Is the methodology feasible and described in sufficient detail to allow the work to be replicable?

Reviewer #1: Yes

Reviewer #2: Yes

Reviewer #3: No

4. Have the authors described where all data underlying the findings will be made available when the study is complete?

Reviewer #1: Yes

Reviewer #2: Yes

Reviewer #3: Yes

5. Is the manuscript presented in an intelligible fashion and written in standard English?

Reviewer #1: Yes

Reviewer #2: Yes

Reviewer #3: Yes

6. Review Comments to the Author

You may also provide optional suggestions and comments to authors that they might find helpful in planning their study.

Reviewer #1: Review of the paper “Assessing the aesthetic attractivity of European butterflies: a web-based survey protocol” (PONE-D-22-26118). This interesting study protocol is part of the project, which aims to describe, quantify, and better understand human subjective preferences in butterfly aesthetics. In the anthropocentric world with human-biased perception, I find this study very useful to develop tools for the effective protection of unpopular non-charismatic invertebrate species in the future. I wonder if and how Authors want to evolve this project to other far less popular invertebrate groups. Please find my specific questions and suggestions for changes below.

Line 91: There is a discrepancy between the real (compare line 60) and the studied number of European butterflies. Please explain why.

Line 167: who and how decided about modifications of natural traits? Human or computer programs? It may have mattered as you want to reveal human-biased preferences.

Table 1: I have a few doubts concerning the selection of species. Some of them are widespread in Europe, but some are typical for particular regions. It may affect the results due to even unconscious human contact with considered species. In consequence, people from regions where butterfly species occur may judge pictures illustrating natural traits as “better”. Are you going to control species range and respondent living place? Or treat it as a potential factor?

Line 201: I’m a little bit surprised by proposing so many negative emotions compared to positive ones.

Lines 204-205: What was the key to selecting listed species?

Line 327: As I understood your study began in May. In April you run the final questionnaire tests excluded from analyses. Is it correct?

Line 333: Please clarify the time frame. Once you give April 2023 and later you mention March 2023.

Line 248: Are you going to process multivariate analysis combining all Sections (i.e. age, sex, nationality, butterfly ranking, morphological features, emotional engagement, and dispositions)?

Line 403: I think an additional limitation may be here language barrier. – questionnaire is available in two languages.

Appendix S2

Line 89: “modified aspect with 100% smaller eyespots” – in this case, it wouldn’t be at all.

Line 104: Some representations of modification look artificial i.e. wing tails of Iphiclides podalirius -like just cutting. Hence my technical question: how was it modified?

Reviewer #2: First of all, I apologize to the authors for my English that is not fluent, but I am not a native English speaker.

Second, maybe I am not the best person to review this kind of manuscript in the sense that the manuscript is highly “artistic” and subjective. Taking the above into account my review is as follows.

After reading the manuscript several times and conducting the “Unveiling test” I am convinced that this protocol should be published in the journal. This kind of research is very rare and the approach of the authors is certainly very original and novel. Despite the fact that I have some doubts as to the proper, clear, and effective design of the protocol I want to stress that I am aware that my point of view (in regard to such an unusual type of study) is certainly distorted by my scientific background. My concerns should rather be a subject of verification by the scientific community, not act as arguments to reject the manuscript, albeit some comments to what is told below could be inserted into the manuscript.

Objections

1) It is not clear to me what type of product as the output of the utilization of the protocol will be provided to the NGOs, environmental associations, authorities, and generally the bodies responsible for nature conservation. Will it be a) a list of the most charismatic species of European butterflies (taxa), b) a list of the morphological characters according to which the umbrella species should be selected, or another kind of document? The authors should be aware of the fact that if the results of your study intend to be useful the message to people working in widely speaking administration has to be precise and clear.

2) Section “Single morphological features”

I am not convinced of the idea of the modification of certain features in the way that they do not reflect the states expressed by real species. One may hypothesize that the species with more dots (case of Erebia) or extremely jagged edges (Polygonia) will appear to be the most charismatic but there are no such species in reality. For me, this is more a purely artistic (or aesthetic) experiment than a strict comparison between more and less charismatic REAL characters.

3) Section “Emotional engagement”

This comment may be affected by my biological education.

I doubt whether any of the specimens of butterflies presented to the tested person will trigger such extreme emotions as those suggested by the authors. In my subjective opinion, none among them will evoke fear or disgust, or at least the percentage of such answers will be very, very low. It is well known that butterflies generally evoke very positive emotions. In such cases how informative will be these exceptionally rare types of answers?

Maybe the better option would be to reduce the possible answers to just a few, with the option of manually typing the answer (emotion).

Moreover, I invited my 18 y.o. son to conduct the test and his comments on this section were similar to mine.

In general, I expect a much less diversified range of emotion than suggested by the authors.

I am really curious to see the results of the experiment, especially concerning this section.

Some technical remarks:

1) Introduction lines 56-59. Aesthetic merits attributed to butterflies are also very strongly correlated with their diurnal activity because also we, human beings are day-active animals. Even more beautiful, night-flying moths are simply unknown to us just because they are nocturnal.

2) Section n 4

l. 295; there is “pale”, should be “darker” ??

l. 300; the experts will also express more positive emotions seeing rare or endangered species.

Reviewer #3: In this study protocol authors describe research design and methods they use to examine how beautiful species of butterflies are to people. I find this topic interesting, however I also think that something is missing in this study to be considered as related to nature conservation. For example, correlations between aesthetic value of species and real preferences of people in nature conservation decision making, their real engagement in actions to protect species. Such preferences can not be studied with questionnaires because people usually declare differently than shown in actions.

An assumption that more beautiful butterflies will be catchier in conservation marketing actions also needs testing, otherwise is speculative. Beauty is not always the case in the charisma of species. For example - is an elephant beautiful? Methods designed by authors for this study can not tell if beauty will have influence on people’s decision making.

The study is already in progress since the middle of 2022 and will be continued till the middle of 2023. I am wondering how the review process, comments and suggestions may help in improving the data collection? In my opinion the review process makes sense for study protocols only if they can be modified and the study can be improved. Here I don’t see this opportunity.

I am also not sure how authors will solve the problem with different backgrounds that may affect points assigned by people, showing how much they like the picture. Effects of colors in the background, the ratio of the background to the body of the butterfly (zoom), posture of the animal as well as the effect of other elements on the picture are not controlled. At this stage of research, it cannot be modified because data collection lasts for half a year already. Therefore in my opinion, there are too many confounding variables that are not controlled in this study.

Another problem I see is that participants are not random thus results may be from a very local population. Origin of respondents is also not diversified in a controlled way. Even if authors would have a sample of 5.000 respondents from Cambridge, results could not be extrapolated on preferences of people in the whole UK, and Europe, if respondents were not random and only from one city.

The definition of beauty used by authors in this study is not clearly stated.

7. PLOS authors have the option to publish the peer review history of their article (what does this mean?). If published, this will include your full peer review and any attached files.

Reviewer #1: No

Reviewer #2: No

Reviewer #3: No

---

## [Author Response · Author response to Decision Letter 0]

22 Dec 2022

Reviewer #1: Review of the paper “Assessing the aesthetic attractivity of European butterflies: a web-based survey protocol” (PONE-D-22-26118). This interesting study protocol is part of the project, which aims to describe, quantify, and better understand human subjective preferences in butterfly aesthetics. In the anthropocentric world with human-biased perception, I find this study very useful to develop tools for the effective protection of unpopular non-charismatic invertebrate species in the future. I wonder if and how Authors want to evolve this project to other far less popular invertebrate groups. Please find my specific questions and suggestions for changes below.

Line 91: There is a discrepancy between the real (compare line 60) and the studied number of European butterflies. Please explain why.

Thank you for pointing out this one, we (unwisely) rounded to 500 the total number of 496. We fixed this part (see line 61).

Line 167: who and how decided about modifications of natural traits? Human or computer programs? It may have mattered as you want to reveal human-biased preferences.

The selected traits were chosen according to specialized literature concerning empirical aesthetics and conservation biology (see: Berlyne, D. E., 1970: “Novelty, complexity, and hedonic value”, Perception & Psychophysics, 8, 279-286; Manesi, Z., Van Lange, P.A.M., Pollet, T.V., 2015: “Butterfly Eyespots: Their Potential Influence on Aesthetic Preferences and Conservation Attitudes”, PLoS ONE, 10(11): e0141433. doi:10.1371/journal.pone.0141433; Gómez-Puerto, G., Rosselló, J., Corradi, G., Acedo-Carmona, C., Munar, E., & Nadal, M., 2018: “Preference for curved contours across cultures”, Psychology of Aesthetics, Creativity, and the Arts, 12, 432-439; Specker, E., Leder, H., Rosenberg, R., Hegelmaier, L., Brinkmann, H., Mikuni, J. & Kawabata, H., 2018: "The universal and automatic association between brightness and positivity", Acta Psychologica, 186, p. 47-53; Leder, H., Tinio, P., Brieber, D., Kröner, T., Jacobsen, T. & Rosenberg, R., 2019: "Symmetry is Not a Universal Law of Beauty", Empirical Studies of the Arts, 37, 1, p. 104-114; Gartus, A., Völker, M. & Leder, H., 2020: "What Experts Appreciate in Patterns: Art Expertise Modulates Preference for Asymmetric and Face-Like Patterns", Symmetry, 12, 5, 27 p., 707). In empirical aesthetics, features such as curved contours, complexity, symmetry, contrast, perspicuity, composition of an image are generally used to “calculate” the level of beauty (or aesthetic attractiveness) of the image. As for the specific case of butterflies, we also referred - to identify the traits to be altered - to Habel, J.C., Gossner, M.M., Schmitt, T. 2021: “Just beautiful?! What determines butterfly species for nature conservation”, Biodivers. Conserv., 30(8): 2481-93, in which four morphological features (average size of forewing; colour of upper wing side; shiny upper wing side; shape of wings) are chosen as main factors in calculating the “index of beauty” of European butterflies. Once completed the selection, we proceeded in creating contrastive pairings aimed at highlighting each altered trait, hence: i) butterfly dimension, ii) colours of the wings contrast intensity, iii) grouping and order of the design patterns of the wings, iv) forewing/hindwing proportion, v) presence or absence of wing eyespots, vi) wing eyespots dimensions, vii) presence or absence of wing tails, viii) wing tail length, ix) smooth or jagged wing edges. The goal is to test if these same traits retain their aesthetic relevance also when applied to our specific case study, i.e. butterflies; no reference to real phenotypes is intended. We made this choice explicit in the revised version of the manuscript at lines 184-188. The drawings’ alterations have been performed by one of the authors through MS PowerPoint.

Table 1: I have a few doubts concerning the selection of species. Some of them are widespread in Europe, but some are typical for particular regions. It may affect the results due to even unconscious human contact with considered species. In consequence, people from regions where butterfly species occur may judge pictures illustrating natural traits as “better”. Are you going to control species range and respondent living place? Or treat it as a potential factor?

We agree that the rate of encounter of a given species might influence how the user judges its aesthetic attractivity. For this reason, the test also collects the country of origin for each participant, as well as their self-assessed knowledge about butterflies and general interest in nature and science. All together, these data will allow control, at least partially, for a possible effect of previous encounters and thus differential knowledge of butterflies for different combinations of user/butterfly species. We would like to say, however, that neither species distribution per se nor its abundance (provided that we could gather this info for all species) are necessarily good proxies of the user's knowledge of butterflies, as this will mostly concern that sector of the test audience which has a significant interest in nature and/or often encounter butterflies. We anticipate that, given that our user recruitment goes well beyond naturalists and people with a strong interest in nature, only a minor part of people responding to the test will be affected, potentially, by this effect. In any case we further highlighted this issue at lines 364-376, and we will consider the potential confounding factors in the following analyses.

Eventually, in light of the highest degree of representation of the actual state of things concerning European butterflies families and tribes, we purposely decided not to make distinctions between widespread and localized items. We shall clarify, however, that at this stage - which is the one the protocol is aimed at reporting - we are only collecting raw data: we are definitely going to take into account the breadth of spread proper to each species during the process of data analysis. We thank the reviewer for having stressed this point.

Line 201: I’m a little bit surprised by proposing so many negative emotions compared to positive ones.

The current balance of emotions was thought of according to the following ranking: 3 negative ones (confusion, disgust, fear), 3 positive ones (joy, cuteness, feeling of challenge - this latter taken as a synonym for “curiosity”), 1 mixed (“awe”, which, depending on contexts and specific situations, may have a positive or negative valence; see Clewis, R. R., Yaden, D. B., Chirico, A., 2021: “Intersections Between Awe and the Sublime: A Preliminary Empirical Study”, Empirical Studies Of The Arts, (N/A): 1-31. [doi:10.1177/0276237421994694]) and “none of these” = indifference or impossibility to describe one’s emotion in the terms suggested above. The ranking has been derived from resources in empirical aesthetics such as: Silvia, P., Brown, E., 2007: “Anger, disgust, and the negative aesthetic emotions: Expanding an appraisal model of aesthetic experience”, Psychology of Aesthetics, Creativity, and the Arts, 1(2), 100-106. DOI:10.1037/1931-3896.1.2.10; Marković, S., 2011: “Components of aesthetic experience: aesthetic fascination, aesthetic appraisal, and aesthetic emotion”, i-Perception, 3, pp. 1-17; Schindler, I., Hosoya, G., Menninghaus, W., Beermann, U., Wagner, V., Eid, M., Scherer, K.R., 2017: “Measuring aesthetic emotions: A review of the literature and a new assessment tool”, PLoS One, https://doi.org/10.1371/journal.pone.0178899; Menninghaus, W., Wagner, V., Wassiliwizky, E., Schindler, I., Hanich, J., Jacobsen, T., Koelsch, S., 2019: “What are aesthetic emotions?” Psychological Review, 126(2), pp. 171-195.

Moreover, the ranking of emotions also kept reference to other classic models such as those by Paul Ekman (Ekman, P., 1992: “An argument for basic emotions”, Cognition and Emotion, 6 (3-4), pp. 196-200) and Jaak Panksepp (Panksepp, J., 1982: “Toward a general psychobiological theory of emotions”, Behavioral and Brain Sciences, 5(3), 407-422 doi:10.1017/S0140525X00012759). In both these models (6 basic emotions, Ekman; 7 primary emotional systems, Panksepp), negative emotions hold a relevant place and are very well represented: Ekman refers to anger, disgust, fear, happiness, sadness and surprise while Panksepp refers to seeking, rage, lust, care, panic/grief, play, fear. To sum up, the list we presented responded as the best compromise we found among: secundary literature concerning aesthetic emotions, empirical studies on emotions based on Charles Darwin’s evolutionary theory and the need to minimize the potential respondent fatigue effect (therefore, we reduced the number of emotions to 7, + “none of these”). As now clearly stated in the protocol, at lines 377-382, we had the chance to measure the impact of negative emotions on the first 500 finished tests during the beta-testing period and we found a percentage of 8,7% of test-takers experiencing negative emotions. Also in light of this tiny yet not negligible outcome, we stuck to the current ranking of emotions.

Lines 204-205: What was the key to selecting listed species?

Thank you for this question as it allowed us to clarify, in the corrections made to the paper, a crucial issue. The selection is aimed at obtaining the greatest degree of diversity and representativity by choosing X species for each subfamily and tribe.

As specified at lines 230-241, once the goal of 2,500 tests will be reached, a second selection of pictures will replace the first catalog. The 25 new species will be chosen according to the same criterion as before. At the end of the 5000 test, thus, this section will have shown 51 species representing at least one item from each tribe/subfamily. This will guarantee adequate representativeness about the diversity of European butterflies.

Line 327: As I understood your study began in May. In April you run the final questionnaire tests excluded from analyses. Is it correct?

Yes, that is correct. We started collecting data on May 1st 2022. All previous data were (and are going to be) excluded from the analyses since they were either coming from internal trials or corrupted by code-bugs and alikes. We clarified this point at line 385.

Line 333: Please clarify the time frame. Once you give April 2023 and later you mention March 2023.

Thank you for pointing out this one as well, we meant to formally conclude the collection of data by March 31st and start analyzing them from April 1st but we took the chance to make things simpler in the paper, now, and merged both the end of the collection and the start of the analyses in April 2023 (see lines 392-393).

Line 248: Are you going to process multivariate analysis combining all Sections (i.e. age, sex, nationality, butterfly ranking, morphological features, emotional engagement, and dispositions)?

Yes, although we did not specify that in a dedicated section in the paper - as the aim of the protocol is not to describe every aspect of our study project but rather to report the steps of its first stage - we are going to process the data collected thanks to the test by the means of multivariate analysis. We gave a hint as to the validity of the data collected, in light of future multi-varied analysis, adding more information about some preliminary results of the project in the section “Preliminary assessment of test appreciation”, at lines 364-382.

Line 403: I think an additional limitation may be here language barrier. – questionnaire is available in two languages.

That is definitely a fair point. However, the current resources and timeframe of the project do not allow the translation into other languages. The choice of using two languages is supported by many previous studies f which produced relevant scientific results and have been carried using even a single or two languages. Indeed, studies as the one by Boso et al. (published in 2021: https://zslpublications.onlinelibrary.wiley.com/doi/full/10.1111/acv.12692) and the one by Langlois et al. (published in 2022: https://journals.plos.org/plosbiology/article?id=10.1371/journal.pbio.3001640), compliant with this limitation, nonetheless producing interesting results and thus having a considerable impact on the scientific debate. In our case we decided to propose our test in English, being the official language of the scientific community, and also in Italian, being our research group’s mother-tongue and the representative language of the area where we expect to exert the highest influence as to the dissemination of our outcomes. We hope to translate the test in other languages in the future. In any case, the interpretation of the external validity of the results will clearly take into account possible language difficulties, also in relation to the country of origin (an information that we can get from our test).

Appendix S2

Line 89: “modified aspect with 100% smaller eyespots” – in this case, it wouldn’t be at all.

Thank you for spotting that typo: it’s actually 50%, our apologies about that.

Line 104: Some representations of modification look artificial i.e. wing tails of Iphiclides podalirius-like just cutting. Hence my technical question: how was it modified?

That is correct: they are artificial; we used MS PowerPoint to alter those traits.

Reviewer #2: First of all, I apologize to the authors for my English that is not fluent, but I am not a native English speaker.

Second, maybe I am not the best person to review this kind of manuscript in the sense that the manuscript is highly “artistic” and subjective. Taking the above into account my review is as follows.

After reading the manuscript several times and conducting the “Unveiling test” I am convinced that this protocol should be published in the journal. This kind of research is very rare and the approach of the authors is certainly very original and novel. Despite the fact that I have some doubts as to the proper, clear, and effective design of the protocol I want to stress that I am aware that my point of view (in regard to such an unusual type of study) is certainly distorted by my scientific background. My concerns should rather be a subject of verification by the scientific community, not act as arguments to reject the manuscript, albeit some comments to what is told below could be inserted into the manuscript.

Objections

1) It is not clear to me what type of product as the output of the utilization of the protocol will be provided to the NGOs, environmental associations, authorities, and generally the bodies responsible for nature conservation. Will it be a) a list of the most charismatic species of European butterflies (taxa), b) a list of the morphological characters according to which the umbrella species should be selected, or another kind of document? The authors should be aware of the fact that if the results of your study intend to be useful the message to people working in widely speaking administration has to be precise and clear.

Thank you for pointing this out. Actually, the protocol here provided does not describe our project in its entirety, but rather the single step of data collection: how we built the online survey in its 5 main sections, how we made our choices as to the selection of butterfly species, which key-features we took into account, how we set up the test for public diffusion and so on. The eventual goal of the overall project will be to introduce the aesthetic trait as a relational trait alongside the functional traits already currently considered in strategies to protect endangered species (see Moretti M, Dias ATC, de Bello F, Altermatt F, Chown SL, Azcárate FM, et al., 2017: “Handbook of protocols for standardized measurement of terrestrial invertebrate functional traits”, Functional Ecology, 31(3), pp. 558–67). Furthermore, we aim at supporting the acquisition of awareness by NGOs about the role of beauty and aesthetic attractiveness in conservation policies.

2) Section “Single morphological features”

I am not convinced of the idea of the modification of certain features in the way that they do not reflect the states expressed by real species. One may hypothesize that the species with more dots (case of Erebia) or extremely jagged edges (Polygonia) will appear to be the most charismatic but there are no such species in reality. For me, this is more a purely artistic (or aesthetic) experiment than a strict comparison between more and less charismatic REAL characters.

It is precisely an aesthetic experiment for the evaluation of the preference expressed in relation to real morphological traits as opposed to modified ones. The modifications are made on purpose because the aim was to create pretty much a whole different species from the real/actual one. We are not comparing actual phenotypes within the natural range of variation, but rather extrapolating traits that are potentially implicated in aesthetic attractiveness by taking them to extremes to identify which among them are most attractive. Just to stress further a point which ought to be clear, the choice of such relevant traits was not arbitrary but driven by the scientific literature and references provided (see: Berlyne, D. E., 1970: “Novelty, complexity, and hedonic value”, Perception & Psychophysics, 8, 279-286; Manesi, Z., Van Lange, P.A.M., Pollet, T.V., 2015: “Butterfly Eyespots: Their Potential Influence on Aesthetic Preferences and Conservation Attitudes”, PLoS ONE, 10(11): e0141433. doi:10.1371/journal.pone.0141433; Gómez-Puerto, G., Rosselló, J., Corradi, G., Acedo-Carmona, C., Munar, E., & Nadal, M., 2018: “Preference for curved contours across cultures”, Psychology of Aesthetics, Creativity, and the Arts, 12, 432-439; Specker, E., Leder, H., Rosenberg, R., Hegelmaier, L., Brinkmann, H., Mikuni, J. & Kawabata, H., 2018: "The universal and automatic association between brightness and positivity", Acta Psychologica, 186, p. 47-53; Leder, H., Tinio, P., Brieber, D., Kröner, T., Jacobsen, T. & Rosenberg, R., 2019: "Symmetry is Not a Universal Law of Beauty", Empirical Studies of the Arts, 37, 1, p. 104-114; Gartus, A., Völker, M. & Leder, H., 2020: "What Experts Appreciate in Patterns: Art Expertise Modulates Preference for Asymmetric and Face-Like Patterns", Symmetry, 12, 5, 27 p., 707). We clarified this point at lines 184-188.

3) Section “Emotional engagement”

This comment may be affected by my biological education.

I doubt whether any of the specimens of butterflies presented to the tested person will trigger such extreme emotions as those suggested by the authors. In my subjective opinion, none among them will evoke fear or disgust, or at least the percentage of such answers will be very, very low. It is well known that butterflies generally evoke very positive emotions. In such cases how informative will be these exceptionally rare types of answers?

Maybe the better option would be to reduce the possible answers to just a few, with the option of manually typing the answer (emotion).

Moreover, I invited my 18 y.o. son to conduct the test and his comments on this section were similar to mine.

In general, I expect a much less diversified range of emotion than suggested by the authors.

I am really curious to see the results of the experiment, especially concerning this section.

Although connoisseurs would hardly experience negative emotions, the preliminary results we gathered show that a tiny yet relevant percentage of test-takers actually expressed disgust and some kind of discomfort when presented with some of the pictures in the test. We updated the protocol, reporting at lines 377-382 the percentage of 8,7% among the total test-takers experiencing negative emotions (with 45 people declaring “disgust”). This given, we consider such an even small percentage of results to still hold legitimate place and relevance in our analyses, hence we would rather keep that choice for the people who should feel like those labels actually comply with the emotion they might be experiencing.

Some technical remarks:

1) Introduction lines 56-59. Aesthetic merits attributed to butterflies are also very strongly correlated with their diurnal activity because also we, human beings are day-active animals. Even more beautiful, night-flying moths are simply unknown to us just because they are nocturnal.

We totally agree with this statement, though this is also precisely the reason why we focused on butterflies instead of moths. By virtue of their daylight activity, butterflies are far more popular within the average population, definitely more eager to get in touch and become accustomed with them than with a number of wonderful, yet rather unpopular moths. Another advantage butterflies are giving to us is: they come in a considerably smaller number of species and families as compared to moths, hence they grant us a significatively wide yet not extremely large and diversified sample. Last but not least, since the test hinges on pictures and, in particular, share-alike copyrighted ones, it makes no surprise to state that the citizen-science repositories we are using to collect the needed images host a number (and a variety) of hi-resolution pictures of butterflies which is far superior to that of moths. This said, we took into account this point and made it more explicit at lines 57-58.

2) Section n 4

l. 295; there is “pale”, should be “darker” ??

This point gave us the chance to fix an important mistake: we erroneously mentioned Hesperiidae where we meant to say Lycaenidae. Sorry about that and thank you for spotting it, you will find due correction at line 333.

l. 300; the experts will also express more positive emotions seeing rare or endangered species.

We agree with the reviewer. While in the previous version we gave it as implicit, we now explicitly refer to the likeliness of this possibility at line 338-339.

Reviewer #3: In this study protocol authors describe research design and methods they use to examine how beautiful species of butterflies are to people. I find this topic interesting, however I also think that something is missing in this study to be considered as related to nature conservation. For example, correlations between aesthetic value of species and real preferences of people in nature conservation decision making, their real engagement in actions to protect species. Such preferences can not be studied with questionnaires because people usually declare differently than shown in actions.

First of all, we shall thank you for your attention as your points touched sensitive topics throughout our study. On a scientific level, we both agree and disagree with your stance concerning questionnaires: it is indisputable that people would act differently compared to what, on average, they might tell they would in reply to a number of direct questions (e.g. Scherpenzeel A.C., Saris, W.E., 1997: “The Validity and Reliability of Survey Questions: A Meta-Analysis of MTMM Studies”, Sociological Methods & Research, 25(3), 341–383, https://doi.org/10.1177/0049124197025003004).

On the other hand, though, it is also true that albeit a questionnaire won’t tell “the truth”, it will nevertheless advocate for stances, attitudes and habits still holding a significant chance to be mirrored in the “real-case” scenarios. On top of that, there are a number of scientific studies carried out on the basis of questionnaires that actually produced highly relevant results for the scientific community (e.g.: Zimmerman, M., Mattia, J. I., 2001: “The Psychiatric Diagnostic Screening Questionnaire: Development, reliability and validity”, Comprehensive Psychiatry, 42(3), pp. 175-189, https://doi.org/10.1053/comp.2001.23126; Skinner, T.C., Howells, L., Greene, S., Edgar, K., McEvilly, A., Johansson, A., 2003: “Development, reliability and validity of the Diabetes Illness Representations Questionnaire: four studies with adolescents”, Diabetic Medicine, 20: 283-289, https://doi.org/10.1046/j.1464-5491.2003.00923.x; Busschaert, C., De Bourdeaudhuij, I., Van Holle, V. et al., 2015: “Reliability and validity of three questionnaires measuring context-specific sedentary behaviour and associated correlates in adolescents, adults and older adults”, Int J Behav Nutr Phys Act 12, 117, https://doi.org/10.1186/s12966-015-0277-2; in spite of their inherent limitations and proved particularly effective on large-scaled studies.

In our view (as supported by other studies, some of them even published by the very PlOS ONE, such as; Garnett S.T., Ainsworth G.B., Zander, K.K., 2018: “Are we choosing the right flagships? The bird species and traits Australians find most attractive”, PLOS ONE 13(6): e0199253. https://doi.org/10.1371/journal.pone.0199253), the interpretative methods and instruments are key to the success or failure of studies conducted this way. The whole protocol presented here is precisely the means by which we aim to explain and motivate each step of the foundational stage of our research.

An assumption that more beautiful butterflies will be catchier in conservation marketing actions also needs testing, otherwise is speculative. Beauty is not always the case in the charisma of species. For example - is an elephant beautiful? Methods designed by authors for this study can not tell if beauty will have influence on people’s decision making.

This study follows an already well established trend in literature concerned with the role of beauty in biological conservation that has been raising, as to its relevance, during the last 20 years with publications such as: Gunnthorsdottir A., 2001: “Physical Attractiveness of an Animal Species as a Decision Factor for its Preservation”, Anthrozoös, 14(4), pp. 204-15; Stokes, D.L., 2007: “Things We Like: Human Preferences among Similar Organisms and Implications for Conservation”, Hum. Ecol., 35(3): pp. 361-9; Frynta, D., Lišková, S., Bültmann, S., Burda, H., 2010: “Being Attractive Brings Advantages: The Case of Parrot Species in Captivity”, PLOS ONE, 5(9) :e12568; Colléony, A., Clayton, S., Couvet, D., Saint Jalme, M., Prévot, A.C., 2017: “Human preferences for species conservation: Animal charisma trumps endangered status”, Biological Conservation, 206, pp.: 263-69; Landová, E., Poláková, P., Rádlová, S., Janovcová, M., Bobek, M., Frynta, D. 2018: “Beauty ranking of mammalian species kept in the Prague Zoo: does beauty of animals increase the respondents’ willingness to protect them?”, Sci Nat., 105 (11-12): 69, doi: 10.1007/s00114-018-1596-3, PMID: 30488357; Langlois, J., Guilhaum, F., Baletaud. F., Casajus, N., DeAlmeida Braga, C., Fleure, ´ V. et al. 2022: “The aesthetic value of reef fishes is globally mismatched to their conservation priorities”, PLoS Biol, 20(6): e3001640. https://doi.org/10.1371/ journal.pbio.3001640. A closer look at this relevant research literature shows, however, a number of gaps and shortcomings.

These and other studies make use of a not sufficiently clear, not adequately grounded notion of beauty (about that, we provided an in-depth definition of beauty in the response to your last comment). Indeed, no reference is to be found to the substantial body of knowledge accumulated so far both in philosophical aesthetics (that is, the branch of philosophy which has to do with the analysis of beauty and the arts) and in empirical aesthetics. Our study attempts at filling this gap, integrating the most advanced research in aesthetics with research in biological conservation. Furthermore, to our best knowledge very few studies on the role of beauty in biological conservation strategies have made use of quantitative methods, which is, instead, what we are relying on. This will allow us to determine which traits (and beauty indeed is one of these traits, although not the only one; empathy, for instance, may also play a role) attribute charisma to a given species and from here - and only at a much later stage of our project - understand how this might reflect on processes of decision making.

The study is already in progress since the middle of 2022 and will be continued till the middle of 2023. I am wondering how the review process, comments and suggestions may help in improving the data collection? In my opinion the review process makes sense for study protocols only if they can be modified and the study can be improved. Here I don’t see this opportunity.

Holding true that a number of tests has already been gathered, such a number, however, is not yet so relevant to obliterate any chance for improvement or modification. Conversely, many aspects can still be refined (i.e.: the very choice of pictures presented to the user) and, most importantly, the reviews to the protocol constitute a crucial means to guide the analyses of the data we will be collecting, taking into account the comments made by fellow researchers and scholars. An example of the interventions we already planned and that are going to be implemented in the nearest future is reported below.

I am also not sure how authors will solve the problem with different backgrounds that may affect points assigned by people, showing how much they like the picture. Effects of colors in the background, the ratio of the background to the body of the butterfly (zoom), posture of the animal as well as the effect of other elements on the picture are not controlled. At this stage of research, it cannot be modified because data collection lasts for half a year already. Therefore in my opinion, there are too many confounding variables that are not controlled in this study.

We agree with the reviewer that context and background information are likely affecting the aesthetic experience (also see Gernot et al., 2014; Leder et al., 2014; Leder et al., 2022) and we were aware this was a difficult choice. Our reasoning went as follows. First, the use of pictures of butterflies coming from entomologist collections/museum specimens or even ad-hoc taken ones would have deprived the test-taker of any feature of a real-life experience. On the other end, arranging real experiences (i.e. on-field experiences) in natural environments was clearly a non-viable option. We thus decided to rely on contributions from iNaturalist (a citizen science platform with millions of users), precisely because of the amateur nature of the pictures there provided, freely taken by non-professionals hence more compliant with the features of actual, real-life experiences. 

This choice obviously inherits a certain amount of variability. In order to limit the effect of the context and other non-controlled variables on “attractiveness”, we acted as follows.

First, we carefully chose each picture (making sure to discard lower-fidelity ones) selecting those representing all the combinations of aesthetically relevant traits, i.e.: i) sex (male or female), ii) dorsal or ventral view, iii) flower or neutral backgrounds, trying our best to keep fair consistence between each background to the extent of not ending up with a too-large spectrum of possibilities. 

Second, we decided to renew the whole database of pictures once half the target number of tests is reached (that is, 2500 completed tests).

Third, relying on a subset of already collected data, we estimated the effect of one of the possible confounding variables (pictures portraying the butterfly on flowers VS casual background) on the aesthetic index. In the preliminary assessment of test appreciation, now included in the protocol at lines 364-376, we demonstrate that the evaluation for the effect of different backgrounds on votes revealed a significant effect with pictures representing a flower in the background receiving higher votes (estimate, 0.165; standard error, 0.0417; z-value, 3.950; p-value <0.001). However, the difference in estimated marginal means in quite low (mean vote without flower 6.42 +- 0.0441 standard error, with flower 6.59 +- 0.0453) and the variance explained by the full model (species+background, conditional R2=0.585) is much higher than the variance explained by background alone (residual R2=0.007). We believe this approach represents a sound trade-off which guarantees an enjoyable and “semi naturalistic” experience while providing substantial information to control and assess the effect of possible confounding variables. We still agree about the influence of the natural setting portrayed in each picture, though we consider that both pre and post testing strategies (analysis) can effectively reduce its impact.

Another problem I see is that participants are not random thus results may be from a very local population. Origin of respondents is also not diversified in a controlled way. Even if authors would have a sample of 5.000 respondents from Cambridge, results could not be extrapolated on preferences of people in the whole UK, and Europe, if respondents were not random and only from one city.

While a test with stratified sampling would be a valuable option, it is currently out of our reach for our project. Moreover, the current literature shows examples of studies (such as the one by Langlois, J., et al., 2022: “The aesthetic value of reef fishes is globally mismatched to their conservation priorities”, PLOS Biology. 2022 Jun 7;20(6):e3001640) which used a snowballing approach and still provided trustable and insightful results. Finally, our scope is not to extrapolate on preferences at a European or large-scale level, but rather to provide a first assessment of the potential biases and (some) of the underlying reasons. We absolutely recognize that further and more structured surveys would be needed in the future to describe the large-scale patterns of aesthetic preferences for butterflies. We believe the current protocol would set a milestone about this.

The definition of beauty used by authors in this study is not clearly stated.

In this article we assume the concept of beauty in the following terms: by ‘beauty’ (e.g. “this butterfly is beautiful") we mean a multifactorial relational property, neither exclusively possessed by the object - here, the butterfly - nor to be understood as the exclusive result of the subject's projection onto the object (therefore relational), which emerges from the subject's first-person experience of the object (in our case, a digital reproduction of the object) and in which perception (the subjects’ perceptual preferences for morphological perceptible features of the object, see Martindale et al., 1988; Zadra et al, 2011; Prinz et al. 2019), emotions (generally positive for beauty; see Berlyne, 1970; Silvia et al., 2007; Schindler et al., 2017; Menninghaus et al., 2019), cognitive information and expertise of the subject about the object (see Gómez-Puerto et al., 2018; Specker et al. 2018), and dispositional variables (the subject’s interests, see Leder et al. 2019; Gartus et al., 2020) all contribute as constituent and integrating factors. In this sense, the various sections of the test correspond item by item to the set of factors involved, in our view, in the human experience of beauty; beauty, as a latent variable, will result from the integration of all these factors (at a later stage of data analysis, not included in this protocol).

---

## [Decision Letter · Decision Letter 1]

14 Feb 2023

PONE-D-22-26118R1Assessing the aesthetic attractivity of European butterflies: a web-based survey protocolPLOS ONE

Dear Dr. Portera,

Thank you for submitting your manuscript to PLOS ONE. After careful consideration, we feel that it has merit but does not fully meet PLOS ONE’s publication criteria as it currently stands. Therefore, we invite you to submit a revised version of the manuscript that addresses the points raised during the review process.

We look forward to receiving your revised manuscript.

Kind regards,

Łukasz Kajtoch, Ph.D.

Academic Editor

PLOS ONE

Additional Editor Comments:

Apologize for the delay in the decision on your manuscript. I have invited all previous three reviewers but only one accepted this invitation, but later have not sent his/her opinion for a long time, and finally, I requested this several times. Meantime, I invited another reviewer and I got one opinion, which points out some other issues in your study, particularly if he/she has some methodological questions. Therefore, I encourage you to revise your manuscript according to that last review (reviewer 4th). 

Reviewers' comments:

Reviewer's Responses to Questions

**Comments to the Author**

1. Does the manuscript provide a valid rationale for the proposed study, with clearly identified and justified research questions?

Reviewer #2: Yes

Reviewer #4: Yes

2. Is the protocol technically sound and planned in a manner that will lead to a meaningful outcome and allow testing the stated hypotheses?

Reviewer #2: Yes

Reviewer #4: Yes

3. Is the methodology feasible and described in sufficient detail to allow the work to be replicable?

Reviewer #2: Yes

Reviewer #4: Yes

4. Have the authors described where all data underlying the findings will be made available when the study is complete?

Reviewer #2: Yes

Reviewer #4: Yes

5. Is the manuscript presented in an intelligible fashion and written in standard English?

Reviewer #2: Yes

Reviewer #4: Yes

6. Review Comments to the Author

You may also provide optional suggestions and comments to authors that they might find helpful in planning their study.

Reviewer #2: I checked the revised version of the manuscript. In my opinion, all comments and remarks raised in my initial review have been scrupulously addressed by the authors. Besides the detailed replies, the expected corrections are made in the manuscript. I also went through the comments from other reviewers and, also in this case, most of the corresponding amendments have been done. Considering the above I recommend accepting the manuscript and qualifying it for the next stages of processing.

Reviewer #4: Assessing the aesthetic attractivity of European butterflies: a web-based survey protocol

Thank you for sending the manuscript for review. Conservation of endangered species is an ever-present topic that is studied from different perspectives. Aesthetic preferences towards animals are a frequently mentioned factor that plays an important role in the planning and implementation of conservation projects in recent years. These projects, including rescue breeding and zoos, cannot do without public support. Finding out which species are positively perceived by people and why therefore provides valuable insights for the future direction of conservation activities. In recent years, attention has been shifting from the charismatic vertebrate groups to less popular groups and also to invertebrates. The focus of the study on butterflies is an appropriate continuation of similar research. While I appreciate the quality of the submitted project, I have several comments and additional questions on the submitted text.

I missed the zoological definition of the study at the beginning of the paper, i.e. which specific taxa your study is concerned with. The specific species are listed in the Appendix, however, a brief mention in the text would have been appropriate.

I have further comments on the description of the methodology, which is unclear in some parts. I do not want to criticise your proposed procedure, I just find some information missing in your description.

In the introduction of the methodology, the reference to the summary of the questions used is marked Appendix S3, but Appendix S1 and S2 are mentioned later in the text. It would be more appropriate to change the numbering.

Section n.2: Ranking

As I understand it, each respondent ranked 9 randomly selected images. Were the butterfly pictures presented one at a time or all at once? Alternatively, was a preview of all the ranked images shown first?

The procedure for selecting images for the dataset is described and detailed in the appendix, however I would appreciate information on how many images in total were used for this section. As there are not the same number of images from each species, it is not possible to simply count.

Section n. 3: Single morphological features

In this section, it is not clear to me what question respondents were asked to choose from a pair of species. Was it the same as in the previous section?

Again, I am missing information on the total number of images tested in this section. There is also no indication of how the drawings of the unmanipulated butterflies were produced or whether they were from any particular source. It would also be useful to indicate what the specific species were selected on. For example, for the analysis of the presence of eye spots, a species that has them present was selected. But were any other parameters taken into account, such as wing colour or shape?

Section n. 4: Emotional engagement

This section details all the species included. However, it is not entirely clear on what basis these species were selected. The last sentence of this section suggests that this was to create a representative sample, but it would have been more appropriate to state this at the beginning. Nevertheless, it should be better explained what the criteria for selecting species were, for example whether the species selected were those that are most abundant in the group or whether this was based on previous studies.

Your manuscript is a study protocol. It begs the question, why are you publishing your research at this stage? Your study is very well designed and promises interesting results, so your procedure is not entirely clear to me. For example, is this a publication needed to complete the study of one of the authors?

I also have a few observations to discuss. You mention that by educating and approaching some non-preferred animals over a long period of time, you can change people's perception of these species. In the case of aesthetic preference, that doesn't seem quite possible to me. If a person likes a blue butterfly because they like the color blue, you can't "force" them to like a red butterfly through education. We can make people aware that the red butterfly exists and that it is important to nature/biodiversity/the planet, but we can't change their personal preference for the color blue. Educational programs should teach people that there are animals they may not like, but they are interesting, important and should pay attention to. This may translate into increased attention when visiting a zoo or deciding whether to donate to a conservation program. But it doesn't change the fact that a person likes a blue butterfly.

I hope my comments help improve your manuscript and I wish you the best of luck with your future research.

7. PLOS authors have the option to publish the peer review history of their article (what does this mean?). If published, this will include your full peer review and any attached files.

Reviewer #2: No

Reviewer #4: **Yes: **Daniel Frynta

---

## [Author Response · Author response to Decision Letter 1]

17 Feb 2023

Reviewer #2: I checked the revised version of the manuscript. In my opinion, all comments and remarks raised in my initial review have been scrupulously addressed by the authors. Besides the detailed replies, the expected corrections are made in the manuscript. I also went through the comments from other reviewers and, also in this case, most of the corresponding amendments have been done. Considering the above I recommend accepting the manuscript and qualifying it for the next stages of processing.

Thank you very much for this kind comment, your findings have been valuable in helping us clarify complex passages which ended up improving our work: we are grateful to you for this.

Reviewer #4: Assessing the aesthetic attractivity of European butterflies: a web-based survey protocol

Thank you for sending the manuscript for review. Conservation of endangered species is an ever-present topic that is studied from different perspectives. Aesthetic preferences towards animals are a frequently mentioned factor that plays an important role in the planning and implementation of conservation projects in recent years. These projects, including rescue breeding and zoos, cannot do without public support. Finding out which species are positively perceived by people and why therefore provides valuable insights for the future direction of conservation activities. In recent years, attention has been shifting from the charismatic vertebrate groups to less popular groups and also to invertebrates. The focus of the study on butterflies is an appropriate continuation of similar research. While I appreciate the quality of the submitted project, I have several comments and additional questions on the submitted text.

I missed the zoological definition of the study at the beginning of the paper, i.e. which specific taxa your study is concerned with. The specific species are listed in the Appendix, however, a brief mention in the text would have been appropriate.

Thank you for your comments and suggestions throughout this review. We agree with the lack of such definition, and we modified the protocol accordingly, now reporting it at lines 56-57.

I have further comments on the description of the methodology, which is unclear in some parts. I do not want to criticise your proposed procedure, I just find some information missing in your description.

In the introduction of the methodology, the reference to the summary of the questions used is marked Appendix S3, but Appendix S1 and S2 are mentioned later in the text. It would be more appropriate to change the numbering.

Thank you for pointing this out, we rearranged the Appendixes’ numbering to reflect the order in which they are cited in the protocol (lines 106, 153, 195, 202, 772, 774, 776).

Section n.2: Ranking

As I understand it, each respondent ranked 9 randomly selected images. Were the butterfly pictures presented one at a time or all at once? Alternatively, was a preview of all the ranked images shown first?

All 9 pictures are displayed at once, then when the user clicks on each picture to rank it, they see that picture enlarged (covering the other previews) with a clickable 0-10 meter to express their choice. Nevertheless, we agree with your point and updated the first image on fig. 1 with the actual sight presented to the user, displaying the 9 previews altogether. We also specified this in the manuscript, at lines 140-142. 

The procedure for selecting images for the dataset is described and detailed in the appendix, however I would appreciate information on how many images in total were used for this section. As there are not the same number of images from each species, it is not possible to simply count.

The total number of images is: 2288 items. We added it in the appendix.

Section n. 3: Single morphological features

In this section, it is not clear to me what question respondents were asked to choose from a pair of species. Was it the same as in the previous section?

In this section we asked the participants to express a simple preference between two modified images (from a set of images described in S3 Appendix). We now specified the procedure at line 176.

Again, I am missing information on the total number of images tested in this section. There is also no indication of how the drawings of the unmanipulated butterflies were produced or whether they were from any particular source. It would also be useful to indicate what the specific species were selected on. For example, for the analysis of the presence of eye spots, a species that has them present was selected. But were any other parameters taken into account, such as wing colour or shape?

The total number of images displayed in this section is 23 and both the drawings of altered and unaltered species were produced using MS PowerPoint. As starting models, we used pictures of real butterflies. Each species selected was chosen because of the perspicuity of their most salient features (e.g. the dimension of eye spots for Aglais io) which, in turn, were chosen according to the referenced entries of scientific literature in experimental aesthetics. No additional parameter other than the most salient feature of each given species was taken into account. 

Section n. 4: Emotional engagement

This section details all the species included. However, it is not entirely clear on what basis these species were selected. The last sentence of this section suggests that this was to create a representative sample, but it would have been more appropriate to state this at the beginning. Nevertheless, it should be better explained what the criteria for selecting species were, for example whether the species selected were those that are most abundant in the group or whether this was based on previous studies.

We agree with the opportunity to mention the purpose of the selection at the beginning, rather than at the end of this section, therefore we revised the protocol at lines 217-221 also explaining better the criteria we took into account for the species selection. We thank you for highlighting this deficiency, as it has made possible to fill a significant gap.

Your manuscript is a study protocol. It begs the question, why are you publishing your research at this stage? Your study is very well designed and promises interesting results, so your procedure is not entirely clear to me. For example, is this a publication needed to complete the study of one of the authors?

We believe that publishing the protocol at this stage could result in several benefits. Given the scope of the project, keeping track of how we set out its very first step (which is that of data collection: how we built the online survey in its 5 main sections, how we made our choices as to the selection of butterfly species, which key-features we took into account, how we set up the test for public diffusion and so on) should add consistency to the whole dissemination process. Moreover, publishing at this stage can help us involve more people interested in our research, fuel debate and scientific discussion on the method adopted and its aims and return a detailed description of the basic elements on which we are grafting a study that we expect to extend to a broader level, which would be harder to render once we move on to the discussion of the results. Lastly, we entrust PLOS’ study protocol publication’s policy as to the opportunity to give due academic credit to the foundations of our work.

I also have a few observations to discuss. You mention that by educating and approaching some non-preferred animals over a long period of time, you can change people's perception of these species. In the case of aesthetic preference, that doesn't seem quite possible to me. If a person likes a blue butterfly because they like the color blue, you can't "force" them to like a red butterfly through education. We can make people aware that the red butterfly exists and that it is important to nature/biodiversity/the planet, but we can't change their personal preference for the color blue. Educational programs should teach people that there are animals they may not like, but they are interesting, important and should pay attention to. This may translate into increased attention when visiting a zoo or deciding whether to donate to a conservation program. But it doesn't change the fact that a person likes a blue butterfly.

Thanks for this observation. The overall aim of the study does not concern the chance to modify people’s tastes, persuading them - on the long run - to alter their appreciation for something: it is rather a matter of broadening the spectrum of what they appreciate and take into account in their aesthetic experiences. To this extent, we adopt a “multi-factorial” notion of aesthetic experience, understood as consisting of perception, emotion, and cognitive information (which is the purpose of the different sections making up the test). As several pieces of research (See: Franklin, M. B., Becklen, R. C., Doyle, C. L., 1993: “The Influence of Titles on How Paintings Are Seen”, Leonardo 26(2), 103–108. https://doi.org/10.2307/1575894; Nodine, C. F., Locher, P. J., Krupinski, E. A., 1993: “The Role of Formal Art Training on Perception and Aesthetic Judgment of Art Compositions”, Leonardo, 26(3), 219–227. https://doi.org/10.2307/1575815; Cupchik, G. C., Shereck, L., Spiegel, S., 1994: “The Effects of Textual Information on Artistic Communication”, Visual Arts Research, 20(1), 62–78. http://www.jstor.org/stable/20715819; Reber, R., Schwarz, N., Winkielman, P., 2004: “Processing Fluency and Aesthetic Pleasure: Is Beauty in the Perceiver’s Processing Experience?”, Personality and Social Psychology Review, 8(4), 364–382. https://doi.org/10.1207/s15327957pspr0804_3; Leder, H., 2001: “Determinants of Preference: When do we like What we Know?”, Empirical Studies of the Arts 19(2), 201–211. https://doi.org/10.2190/5TAE-E5CV-XJAL-3885; Leder, H., Carbon C.C., Ripsas, A-I., 2006: “Entitling art: Influence of title information on understanding and appreciation of paintings”, Acta Psychologica 121 (2), 176-198.; Mari, E.; Quaglieri, A.; Lausi, G.; Boccia, M.; Pizzo, A.; Baldi, M.; Barchielli, B.; Burrai, J.; Piccardi, L.; Giannini, A.M. Fostering the Aesthetic Pleasure, 2021: “The Effect of Verbal Description on Aesthetic Appreciation of Ambiguous and Unambiguous Artworks”, Behav. Sci. 11, 144. https://doi.org/10.3390/bs11110144) show, what we know (about an object or a living being) affects the kind of emotional connection we establish with it and modifies (at least to a certain extent) the way it appears to us. Therefore, the overall aim of the study is not to “force” people to change their perception – under the important premise that basic perceptual data are not the only component of the aesthetic. It is rather a matter of broadening the spectrum of what we like and of the aesthetic categories that we apply to the natural world (not just “beautiful” and “pleasurable”, but also “sublime”, “eerie”, “surprising”, “diverse”, “magnificent” etc); it is a matter of becoming aware of our (often implicit) preferences and biases and, by means of knowledge and increasing expertise, to mold and shape them in order to make them broader and more inclusive.

I hope my comments help improve your manuscript and I wish you the best of luck with your future research.

They surely did and we are grateful for these important inputs. Thank you for your dedication and good wishes.

---

## [Decision Letter · Decision Letter 2]

7 Mar 2023

Assessing the aesthetic attractivity of European butterflies: a web-based survey protocol

PONE-D-22-26118R2

Dear Dr. Portera,

We’re pleased to inform you that your manuscript has been judged scientifically suitable for publication and will be formally accepted for publication once it meets all outstanding technical requirements.

Kind regards,

Łukasz Kajtoch, Ph.D.

Academic Editor

PLOS ONE

Additional Editor Comments (optional):

Reviewers' comments:

Reviewer's Responses to Questions

**Comments to the Author**

1. Does the manuscript provide a valid rationale for the proposed study, with clearly identified and justified research questions?

Reviewer #4: Yes

2. Is the protocol technically sound and planned in a manner that will lead to a meaningful outcome and allow testing the stated hypotheses?

Reviewer #4: Yes

3. Is the methodology feasible and described in sufficient detail to allow the work to be replicable?

Reviewer #4: Yes

4. Have the authors described where all data underlying the findings will be made available when the study is complete?

Reviewer #4: Yes

5. Is the manuscript presented in an intelligible fashion and written in standard English?

Reviewer #4: Yes

6. Review Comments to the Author

You may also provide optional suggestions and comments to authors that they might find helpful in planning their study.

Reviewer #4: The authors properly responded to all my questions and adequately improved the manuscript according to my suggestions. The manuscript is acceptable in its present form. Of course, it would be better to publish results of the completed study instead of this conceptual and methodological outline.

7. PLOS authors have the option to publish the peer review history of their article (what does this mean?). If published, this will include your full peer review and any attached files.

Reviewer #4: **Yes: **Daniel Frynta

---

## [Editor Report · Acceptance letter]

26 Apr 2023

PONE-D-22-26118R2 

Assessing the aesthetic attractivity of European butterflies: a web-based survey protocol. 

Dear Dr. Portera:

I'm pleased to inform you that your manuscript has been deemed suitable for publication in PLOS ONE. Congratulations! Your manuscript is now with our production department. 

Kind regards, 

on behalf of

dr hab. Łukasz Kajtoch 

Academic Editor

PLOS ONE